# Gut Microbiota Restores Central Neuropeptide Deficits in Germ-Free Mice

**DOI:** 10.3390/ijms231911756

**Published:** 2022-10-04

**Authors:** Sevag Hamamah, Mihai Covasa

**Affiliations:** 1Department of Basic Medical Sciences, College of Osteopathic Medicine, Western University of Health Sciences, Pomona, CA 91766, USA; 2Department of Medicine and Biomedical Sciences, College of Medicine and Biological Science, University of Suceava, 7200229 Suceava, Romania

**Keywords:** leptin, Neuropeptide Y, Agouti-related peptide, ghrelin, germ-free, gut bacteria

## Abstract

Recent work has demonstrated the ability of the gut microbiota (GM) to alter the expression and release of gut peptides that control appetite and regulate energy homeostasis. However, little is known about the neuronal response of these hormones in germ-free (GF) animals, especially leptin, which is strikingly low in these animals. Therefore, we aimed to determine the response to exogenous leptin in GF mice as compared to conventionally raised (CONV-R) mice. Specifically, we injected and measured serum leptin in both GF and CONV-R mice and measured expression of orexigenic and anorexigenic peptides NPY, AgRP, POMC, and CART in the hypothalamus and hindbrain to examine whether the GM has an impact on central nervous system regulation of energy homeostasis. We found that GF mice had a significant increase in hypothalamic NPY and AgRP mRNA expression and a decrease in hindbrain NPY and AgRP mRNA, while mRNA expression of POMC and CART remained unchanged. Administration of leptin normalized circulating levels of leptin, GLP-1, PYY, and ghrelin, all of which were significantly decreased in GF mice. Finally, brief conventionalization of GF mice for 10 days restored the deficits in hypothalamic and hindbrain neuropeptides present in GF animals. Taken together, these results show that the GM regulates hypothalamic and hindbrain orexigenic/anorexigenic neuropeptide expression. This is in line with the role of gut microbiota in lipid metabolism and fat deposition that may contribute to excess fat in conventionalized animals under high feeding condition.

## 1. Introduction

Within the human gastrointestinal tract lives a vast microbial entity known as the gut microbiota. While predominantly providing metabolic and immunological benefits to its host, several lines of research suggest that the gut microbiota has a role in maintaining host energy homeostasis as well as being a contributing factor in the devolvement of obesity. Specifically, compositional changes in the gut microbiota, either in microbial diversity or a reduction in numbers, have been linked to the devolvement of obesity and associated metabolic disorders [1]. This has been readily demonstrated in obese humans and rodents with reduced microbial diversity and or abundance of certain microbial phyla and genera that increase the capacity to absorb energy [2].

Germ-free mice (GF) have been at the forefront in the exploration of the link between gut microbiota and host metabolism. Specifically, GF animals represent ‘knockout’ animals that lack morphological and physiological characteristics associated with supporting intestinal microbiota. Historically, GF mice have been shown to be resistant to obesity when placed on a high-fat diet and have lower levels of adiposity and leptin when compared to their conventionally raised counterparts who have an intact gut microbiota [3,4]; however, several more recent studies do not support this notion [5,6]. For example, both GF and conventional mice gained weight after high-fat feeding, indicating that gut microbiota may not be the necessary factor underlying obesity resistance [5]. Notwithstanding, the impact of gut microbiota on hormones influencing satiety and hunger and its underlying mechanisms are not completely known. As such, in this study we investigate the metabolic participation of gut microbiota by exposing isolated GF mice to normal laboratory conditions outside the isolators, therefore inducing a shift in gut microbial composition.

Previous work showed that expression of nutrient receptors and transporters in GF animals is significantly altered throughout the GI tract [7,8]. The activation of nutrient receptors leads to the release of intestinal satiety peptides, such as glucagon-like peptide-1 (GLP-1) and peptide YY (PYY). However, expression of these peptides is decreased in GF mice [9] while conventionalization or prebiotic treatment increase circulating GLP-1 and PYY with a concomitant decrease in plasma ghrelin [10]. Alterations in nutrient sensing and peptide hormone expression due to lack of microbiota may result in altered fat intake in GF animals [9,11]. For example, the nutrient receptor, GPR41, that is expressed in adipocytes and colonic epithelium, is activated by SCFA. GPR41-deficient GF mice are associated with a reduction in PYY, indicating a dependent role of SCFA-producing gut microbiota in energy harvest [7]. Additionally, secretion of ghrelin, an orexigenic peptide, may be modulated by gut microbiota [12]. As such, GF mice fed high-fat diets had a 10-fold increase in ghrelin, hyperphagia, and obesity as compared to controls [13]. These effects are mediated by increased production of acetate, a 2 carbon SCFA, further indicating the importance of gut microbiota in gut peptide secretion [14]. 

Furthermore, initial studies on brain areas and factors regulating food intake and body weight point to leptin, a non-glycosylated peptide hormone, as one of the paramount peripheral signaling molecules that relays information regarding energy status of the organism to the CNS [15]. The lack of leptin or the leptin receptor results in obesity due to the combined effects of hyperphagia and decreased energy expenditure [16]. Leptin is secreted by adipocytes, in relation to the amount of body fat, and acts as a signal of energy sufficiency. During times of nutritional abundance, adequate leptin levels suppress feeding and allow for energy expenditure. Conversely, starvation and low leptin levels increase the drive to feed and trigger neuroendocrine responses that limit energy expenditure [17]. Leptin influences the arcuate nucleus (ARC) of the hypothalamus by binding to the leptin receptors on neurons that express anorexigenic and orexigenic peptides. On the one hand, leptin activates anorexigenic neurons that express proopiomelanocortin (POMC) and cocaine- and amphetamine-regulated transcript (CART), thereby decreasing food intake and favoring weight loss. On the other hand, the hormone inhibits neurons that express orexigenic peptides, neuropeptide-Y (NPY), and agouti-related protein (AgRP), thereby increasing food intake and weight gain [15]. The balance between the neural circuitry activity of these co-expressing neurons is critical to body weight regulation. The application of high-fat diet-induced obesity models has shown that the microbiota contributes to obesity by increasing energy extraction, promoting inflammation, and altering lipogenic and adipogenic enzymes [18,19]. However, little is known about the neuronal response of these hormones in GF animals, especially the role of leptin, which has been found to be significantly reduced.

Therefore, our study aimed at examining the effects of gut microbiota in energy homeostasis and contribution to obesity development, specifically in leptin-related central pathways. To do this, we first examined changes in expression of orexigenic and anorexigenic peptides NPY, AgRP, POMC, and CART in the hypothalamus and hindbrain of GF and CONV-R animals and whether restoration of gut microbiota through bacterial colonization restored these changes. In addition, we determined leptin, GLP-1, PYY, and ghrelin responses in GF and CONV-R mice challenged with exogenous leptin, a peptide that is well known for its effects on hypothalamic and hindbrain circuitries, influencing food intake and energy balance.

## 2. Materials and Methods

### 2.1. Animals and Diet

Germ-free (GF, *n* = 17) mice from in-house GF colonies and conventionally raised (CONV-R, *n* = 17) C57/B6J 5-week-old male mice from Charles River Laboratories, France, were used in the experiments. All animals were individually housed in a temperature-controlled vivarium with a 12:12 light/dark cycle in polycarbonate cages with a stainless steel metal bottom. The GF and CONV-R groups were housed separately using two Trexler-type isolators (Igenia, Montreuil, France). Sterility of the germ-free isolator was verified weekly by microscopic examination and cultures of freshly voided fecal samples. Both groups of mice received similar autoclaved, deionized water and irradiated standard rodent chow (Safe Diets, Brussels, Belgium) ad libitum, unless noted otherwise. Mice were allowed a minimum of one-week acclimation before experimental manipulations began. Body weights of mice were also recorded daily for 5 days after acclimation. All procedures were carried out in accordance with the European Guidelines for the care and use of laboratory animals.

### 2.2. Experimental Protocol

Before experiments commenced, 5 GF and 5 CONV-R mice were used to establish baseline readings for the subsequent experiments. Both the GF and CONV-R animals underwent a 5-h fast (0900–1400) before being sacrificed. Their blood and brains were collected and tested for mRNA levels of hypothalamic and dorsal hindbrain orexigenic and anorexigenic peptides NPY, AgRP, POMC, and CART. Additionally, their epididymal fat pads were removed, weighed, and adiposity index calculated (total fat/body weight × 100). The remaining GF and CONV-R mice (*n* = 12/group) were separated into two cohorts, each containing 6 mice from both GF and CONV-R groups. That is, cohort 1 had 6 GF and 6 CONV-R mice, while cohort 2 also had 6 GF and 6 CONV-R mice. Cohort 1 was used for the leptin response study in which mice received a series of IP leptin injections after which they were sacrificed. Cohort 2 was used for the conventionalization study where GF mice were conventionalized (CV) via exposure to normal laboratory conditions outside of the isolators. This method of conventionalization should not be confused with transplantation of gut microbes from a conventional host to a GF host and was confirmed via microscopic analyses and feces bacterial cultures. After 10 days, the cohort was sacrificed, and their brain and blood were removed for a similar analysis to the originally sacrificed mice (mRNA levels of orexigenic and anorexigenic peptides). Animals were killed through decapitation, and their brain tissue was excised and stored in AllProtect Tissue Reagent (Qiagen, Courtaboeuf, France) at 2 °C pending RNA extraction. 

### 2.3. Leptin Treatment

Recombinant murine leptin (Sigma) was dissolved in 0.9% saline solution and sterile filtered through a 0.22 µm filter. Both GF and CONV-R mice were injected intraperitoneally (IP) with 4µg/g of leptin or saline vehicle twice a day for 3 days after a 17-h (1700–1000) fast. The animals were sacrificed 120 min at 1200 after the final injection.

### 2.4. Plasma Analysis

Blood collection was achieved via decapitation, and fresh trunk blood was collected in EDTA-coated tubes containing 35 µL/mL aprotinin (Sigma, St. Quentin Fallavier, France), 20 µL/mL pefabloc (Sigma, St. Quentin Fallavier, France), and 10 µL/mL DPP-4 inhibitor (Millipore, Molsheim, France), centrifuged at 3500× *g* at 4 °C, plasma aliquoted, snap-frozen, and stored at −80 °C for further analysis. Blood from GF and CV mice receiving exogenous leptin was tested for concentrations of GLP-1, PYY, ghrelin, and leptin using Enzyme-Linked Immunosorbent Assays (Millipore, Molsheim, France) according to manufacturer’s instructions.

### 2.5. RNA Extraction PCR (qRT-PCR) Analysis

Hypothalamic and hindbrain tissue was lysed and homogenized with a rotor homogenizer. RNA was extracted using TRIzol (Invitrogen, Saint Aubin, France), and the resulting RNA was quantified with NanoDrop (Thermo Scientific, Illkirch, France). Then, 10 μg of RNA was reverse transcribed into 100 μL cDNA using a high-capacity cDNA kit (Applied Biosystems, Courtaboeuf, France). Subsequent cDNA was diluted 5-fold and qPCR performed in a reaction volume of 20 µL using an ABI Prism 7700 thermal cycler (Applied Biosystems, Courtaboeuf, France). All cDNA samples were run in triplicate. Transcription levels of AgRP, NPY, POMC, and CART were quantified using inventoried Taqman Gene Expression Assays and Gene Expression Master Mix (Applied Biosystems, Courtaboeuf, France). Relative mRNA expression was quantified using the 2^−ΔΔCT^ method with β-actin as internal control.

### 2.6. Statistical Analysis

All statistics were analyzed by Statistical Analysis Software (SAS 9.2, SAS Institute Inc., Cary, NC, USA), and data are expressed as means ± SEM. Bi-weekly average body weights were analyzed with repeated measure Analysis of Variance (mANOVA), with post hoc Bonferroni adjustment. Plasma leptin, GLP-1, PYY, and ghrelin levels were analyzed by two-way (group × treatment) ANOVA, with Bonferroni post-hoc tests. Significance was considered at a *p* < 0.05 for all tests. 

## 3. Results

### 3.1. Body Weight, Adiposity, Hindbrain, and Hypothalamic Neuropeptide Expression

There was no difference in the body weights of GF and CONV-R mice; however, GF mice had significantly less adipose tissue compared to their CONV-R counterparts. CONV-R controls (*n* = 5) and GF mice (*n* = 5) had similar body weights averaging around 20 g. However, GF mice had significantly less adipose tissue than the CONV-R mice (Figure 1). GF mice (*n* = 5) had decreased mRNA expression of NPY and AgRP relative to CONV-R mice (*n* = 5) in the hindbrain, while having increased NPY and AgRP mRNA expression in the hypothalamus. POMC and CART mRNA expression in both neuronal regions did not significantly differ between the two groups (Figure 2 and Figure 3).

### 3.2. Neuronal and Gut Peptide Response Following Leptin Administration

GF mice receiving saline had significantly lower levels of GLP-1, PYY, ghrelin, and leptin when compared to CONV-R mice given saline (Figure 4). Moreover, GF mice given IP leptin had significantly lower levels of circulating leptin than CONV-R mice given IP leptin (*p* < 0.05). There was a significant increase in circulating leptin levels between leptin and saline treatment in both GF and CONV-R mice as well as between leptin-treated animals. Similarly, GLP-1 and PYY levels were increased in GF mice given leptin compared to saline-treated mice. Notably, and unlike the other gut peptide responses, ghrelin levels between mice given leptin did not significantly differ between groups. Interestingly, while not statistically significant, CONV-R mice given leptin had less circulating ghrelin than those receiving saline. This may account for the normalization of ghrelin levels between CONV-R and GF given leptin (Figure 4).

### 3.3. Conventionalization Restores Neuropeptide Deficits

Conventionalization of GF mice restored levels of orexigenic neuropeptides in the hindbrain (Figure 5). There was no significant difference in hindbrain and hypothalamic neuropeptide expression between CONV-R and conventionalized (CV) mice except for AgRP in the hypothalamus (Figure 6). In the hypothalamus, CV NPY levels were decreased to similar levels of the CONV-R, while AgRP significantly decreased to below half of the CONV-R concentrations after conventionalization in CV mice (Figure 3 and Figure 6). Again, as with GF mice, no difference was observed in POMC or CART mRNA expression within either brain region (Figure 5 and Figure 6).

## 4. Discussion

In the present study, we showed that leptin administration normalized gut peptides and leptin expression in GF mice and that conventionalization restored the deficits in central neuropeptides present in GF mice. There have been many proposed hypotheses to explain reduced adipose stores in GF animals. Some studies have shown that GF mice are not generally protected from obesity, rather the dietary components and not the macronutrient composition determines the extent of protection [4]. Further, host species differ in the composition and relative abundance of gut microbial species, which can influence absorption of nutrients and retainment of fats. When comparing control and diet-induced obesity mice, Bagarolli et al. found that regardless of changes in the gut microbiota, there was a greater increase in the prevalence of Bacteroidetes and a decrease in the Firmicutes and Actinobacteria phylum in obese mice [20]. While these findings might disagree with other animals studies, the Bacteroidetes/Firmicute phylum ratio in feces was increased while weight was reduced either by a fat-restricted or carbohydrate-restricted diet in humans [21], supporting the role of gut microbiota in promoting weight gain. However, absence of gut microbiota preserved adiposity levels in GF F344 rat model, contrary to findings reported in C57B1/6J mice [22]. Further, GF rats lacking gut microbiota showed altered expression of intestinal proteins and subsequent hepatic energy homeostasis. It was suggested that adiposity in GF rats may result from enhanced local adipose lipogenic activity through downregulation of fasting-induced adipose factor (FIAF) in adipocytes, as opposed to increased FIAF in the GF mice model [22]. Our current results continue to support findings of decreased adiposity in GF C57B1/6J mice model [9,22], indicating an important distinction between host species in regard to adiposity. Furthermore, although evidence has supported the role of gut microbiota in the onset of obesity, recent studies have refuted the notion that they are an indispensable factor [5,23]. As such, it is important to note that other factors besides gut microbiota normalize adiposity and body weight homeostasis even in germ-free mice. However, previous studies have not yet elucidated the role of gut microbiota in restoration of central neuropeptide deficits, a possible mechanism by which gut microbiota influence obesity. 

Our findings show that GF mice had a significant increase in hypothalamic NPY and AgRP mRNA expression and a decrease in hindbrain NPY and AgRP mRNA, while mRNA expression of POMC and CART remained unchanged. This is in line with the findings of Schéle et al., who also showed higher hypothalamic NPY and AgRP expression in the hypothalamus of GF compared to CONV-R mice [2]. However, these authors found no difference in brainstem expression of NPY and AgRP, while we show decreased expression in the hindbrain of GF mice. The hypothalamus and hindbrain, which includes the brainstem and dorsal vagal complex, are brain regions known for feeding, energy, and body fat regulation [24]. More specifically, the arcuate nucleus (ARC) of the hypothalamus houses orexigenic or appetite-increasing neuropeptides, including NPY and AgRP neurons, which are influenced by various feedback mechanisms [25]. As such, decreased hypothalamic NPY and AgRP levels in CONV-R mice may be due to compensatory mechanisms or decreased feedback due to differences in body fat as suggested by Schéle et al. [2]. It is well known that neuronal control of NPY and AgRP are interconnected, with recent findings demonstrating that continuous signaling via NPY increased feeding behavior and activated AgRP hypothalamic neurons [26]. Additionally, NPY is present in enteric neurons, which allows for interactions with the gut bacterial-derived neuroactive metabolites that facilitate gut–brain communication [27,28]. SCFA production by gut microbes increases vagal afferent activity by directly acting on intestinal vagal terminals within enteric neurons [29,30]. Interestingly, acetate, an SCFA, decreases appetite by suppressing NPY and AgRP serum levels through reductions of ARC GABAergic neurotransmission [31]. These findings are supported by studies showing reduced NPY serum levels following probiotic treatment [32]. Taken together, it is plausible that the lack of gut microbes in GF mice and the resulting decreases in circulating neuroactive metabolites led to decreased suppression of orexigenic neuropeptides as seen by increased NPY and AgRP mRNA levels in GF mice.

We found no statistically significant difference in expression of POMC and CART in the hypothalamus or the hindbrain between GF and CONV-R mice. This is in contrast with results from Schéle et al., who found increased POMC and CART in the hypothalamus, but not in the brainstem, in CONV-R compared with GF mice [2]. A key difference between our study and the study of Schéle et al. is that the mice in the present study were subjected to a 5-h fast prior to measurement of POMC and CART mRNA levels, while the mice in the latter study were not fasted. It is thus possible that the difference in eating patterns at the time of measurement altered the availability of food to the colon, where a majority of gut microbiota reside and produce SCFA. As such, varying amounts of neuroactive SCFAs may have been present at time of measurement, which can be attributed, to some extent, to a fasting or non-fasting state. 

Administration of leptin normalized circulating levels of leptin, GLP-1, PYY, and ghrelin, all of which were significantly decreased in GF mice. CONV-R controls had significantly higher levels of circulating gut peptides than their GF counterparts, measured after saline injection. As expected, CONV-R mice had more fat mass and higher levels of leptin than the GF mice after saline and leptin injections. These results are in line with our previous studies demonstrating that decreased GLP-1 and PYY levels in GF mice may be associated with increase in total energy intake [22]. We saw that leptin treatment increased circulating GLP-1 and PYY in GF mice to levels similar of the controls, whereas the excess leptin did not significantly change the amount of GLP-1 nor PYY released in the CONV-R mice. GLP-1 and PYY secretion by intestinal cells is stimulated postprandially and by SCFA in the gut [33]; therefore, excess leptin may have created a similar postprandial response in the GF mice independent of the gut microbiota. Input via vagal afferents from the gut to GLP-1-producing neurons may be modulated by the gut microbiota as evidence of reduced GLP-1 precursor *Gcg* in the brainstem [2]. This decreased *Gcg* expression may contribute to increased host fat mass, which could explain why conventionally raised mice maintain more body fat than GF mice. 

Gut microbiota may have contributed to the unchanged levels of peptides after excess leptin exposure. Our results suggest that leptin may upregulate anorexigenic peptides in the brain and or GI tract, thereby enforcing a greater response in reducing fat gain; however, this mechanism may be halted or suppressed by the gut microbiota. Therefore, gut microbiota may contribute to weight gain and leptin resistance by inhibiting satiety peptide release. There are few mechanisms proposed in the literature that can explain the observed increase in leptin responses of GF mice in response to the exogenous leptin. One such mechanism is through the activity of suppressor of cytokine signaling 3 (SOCS3), which is an inhibitor of STAT3 and leptin signaling [34,35]. The STAT3 gene has been proven to be an important factor in leptin signaling, as deletion of the STAT3 gene in the CNS induces an obese phenotype associated with lower mRNA levels of POMC [36]. SOCS3 is increased in CONV raised mice in comparison to GF mice [2]. One would expect an increase in the protein expression of hypothalamic STAT3 and pERK/ERK in response to peripheral leptin in GF mice; therefore, leptin responsiveness may be hindered by the presence of gut microbiota. Thus, an increased expression of STAT3 and leptin responses may be due to decreased SOCS3 in GF mice. In the short term, increased expression of hypothalamic STAT3 and pERK/ERK can be beneficial by increasing leptin concentrations and blunting the over-feeding response. However, increased leptin expression as measured by STAT3 and pERK/ERK activity can confer leptin resistance [37], though it has been shown that the gut microbiota effect on leptin response varies with diet composition.

Further, the differences in hypothalamic inflammation between CONV and GF may play a role in leptin responsiveness [38]. For example, induction of hypothalamic inflammation via increased microglial activity has been correlated with leptin resistance [39]. Microglia serve as important mediators of neuroinflammation [40], and microglial cells in GF are defective and immature, resulting in inability to respond to key inflammatory signals [41]. Microbiota-derived fermentation products, such as SCFA, promoted maturation of microglia through activation of free fatty acid receptor 2 (FFAR2). FFAR2-deficient mice also showed defective microglia, indicating a key role of microbial metabolites in promoting hypothalamic inflammation. Therefore, GF mice would be expected to have decreased ability to promote neuroinflammation and increased leptin response, which would be consistent with increased STAT3 and ERK expression in response to exogenous leptin.

Additionally, probiotic treatment in high-fat diet-induced obese mice (DIO) improved leptin sensitivity through increased JAK2 and STAT3 phosphorylation in the hypothalamus [20]. Prior studies have also showed that specific gut microbiota modulation and presence of beneficial species into the gut, as measured through an increased Bacteroidetes/Firmicutes ratio, improves leptin response [42]. More specifically, species that are common in probiotics, such as *Bifidobacterium* and *Lactobacillus,* had beneficial effects on leptin sensitivity and were found to be decreased overall in leptin-deficient mice [43,44]. Therefore, it appears that the effects of gut microbiota on leptin responses is dependent on specific species as well as gut enterotypes, which can be altered by depletion or addition of probiotics. Still, the overall presence of gut microbiota versus the lack thereof in GF mice as seen through reported differences in SOCS3 and hypothalamic inflammation offers a plausible mechanism for the increased leptin sensitivity.

We also assessed ghrelin levels in response to peripheral leptin injection. Ghrelin has been a target for obesity and related disease treatment, as it is the only known hormone to stimulate appetite and food intake [45]. Leptin is known to inhibit gastric cell secretion of ghrelin and suppress expression of ghrelin receptors in the NPY system in the arcuate nucleus [46]. In our study, GF ghrelin levels remained unchanged, while the CONV-R ghrelin levels decreased in response to injected leptin. Therefore, our findings suggest that processes involving the gut microbiota contribute to leptin-induced suppression of ghrelin, though the mechanisms that mediate these processes are not yet clear. Several studies have targeted the known ghrelin-producing gut bacteria in hopes to reduce weight gain; however, the results point to Bacteroidetes/Firmicutes ratio as a more important modulator of ghrelin production than any single bacterial species [12,21]. 

Short-term conventionalization of GF mice (10 days) restored the deficits in hypothalamic and hindbrain neuropeptides present in GF animals. Specifically, we found no significant difference in hindbrain or hypothalamic NPY levels between CV and CONV-R mice, while AgRP levels were decreased only in the hypothalamus of CV animals compared with CONV-R mice and were normalized in the hindbrain. POMC and CART levels were no different in CV mice to CONV-R mice, similar to what was observed when comparing GF and CONV-R mice. These results show that gut microbiota regulates hypothalamic and hindbrain orexigenic and anorexigenic neuropeptide expression independent of obesity. This is in agreement with other findings, albeit in a piglet model, showing changes in hypothalamic neurotransmisison following depletion of gut microbiota via antibiotic treatment [47]. In our study, GF mice had increased levels of hypothalamic NPY and AgRP, which were subsequently decreased to normal and below normal levels, respectively, after conventionalization. As mentioned above, SCFA can suppress NPY and AgRP levels through reductions of GABAergic neurotransmission in the arcuate nucleus of the hypothalamus [31], supporting the findings in the current study. 

Lastly, our findings show that orexigenic neuropeptides were increased to normal levels in the hindbrain after conventionalization. The nucleus tractus solitarius (NTS) is a component of the medulla oblongata that plays an important role in the modulation of orexigenic peptides and long-term control of food intake via vagal input from the gut [48]. Therefore, it is possible that neuroactive metabolites produced by gut microbiota can stimulate vagal afferents to activate NTS- and NPY-expressing neurons. Additionally, NTS neurons project densely to the hypothalamus to modulate feeding through signaling via AgRP and POMC neurons [49]. Activation of brainstem nuclei, including NTS neurons involved in promoting satiety, has been shown to inhibit orexigenic neuropeptides like AgRP in the hypothalamus [50]. Thus, the significant decrease in hypothalamic AgRP after conventionalization observed in our study can be due to gut-stimulated signaling via a vagal–NTS pathway that promotes strong attenuation of AgRP neuronal activity in the hypothalamus [48].

## 5. Conclusions

In conclusion, our study provides insight into the role of gut microbiota on leptin responsiveness and shows that it regulates hypothalamic and hindbrain orexigenic/anorexigenic neuropeptide expression that play an important role in regulation of energy and accrual of body adiposity. This is in accordance with the known role of gut microbiota in lipid metabolism and fat deposition that may contribute to excess body fat in regular animals under high calorie feeding.

Notwithstanding the significant importance of these findings, our study has several limitations. First, we did not measure food intake that might have helped in corroborating neuronal changes in GF mice with behavioral and phenotypical data. Second, our study would have benefited from the inclusion of female mice, given the sex-dependent differences in the gut microbiota profile between sexes. Additionally, our study evaluated the overall effect of gut microbiota on neuropeptides that modulate energy intake and appetite. Still, the abundance or lack thereof of specific bacterial phyla and genera can confer varying effects on these same neuropeptides. Several studies have shown that low gut species richness is associated with higher adiposity, insulin resistance, and dyslipidemia [for review see [51,52]]. The loss of gut microbial diversity has been proposed as a hallmark of Western societies when compared with rural areas around the world. This concept has been explored in mice, and research demonstrates that a Western-like diet with reduced fiber maintained over generations results in an eventual loss of bacterial diversity. Therefore, studying the effects of individual phyla on these neuropeptides can be an important future step in identifying key microbial species that can be used in therapeutic modalities to restore orexigenic and anorexigenic neuropeptide levels.

## Figures and Tables

**Figure 1 ijms-23-11756-f001:**
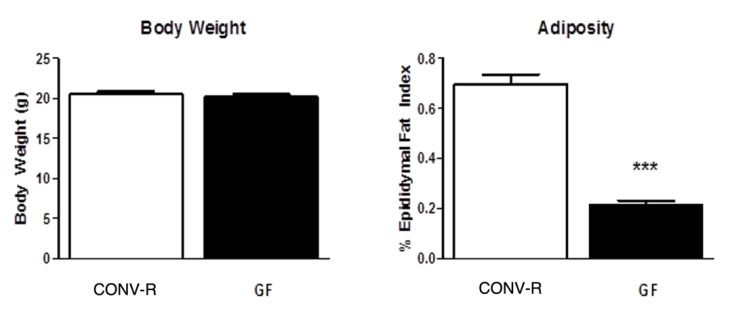
Body weight and adiposity of GF and CONV-R mice after a 5-h fast. Data are expressed as means ± SEM. *** denotes statistical difference between GF and CONV-R, *** *p* < 0.001.

**Figure 2 ijms-23-11756-f002:**
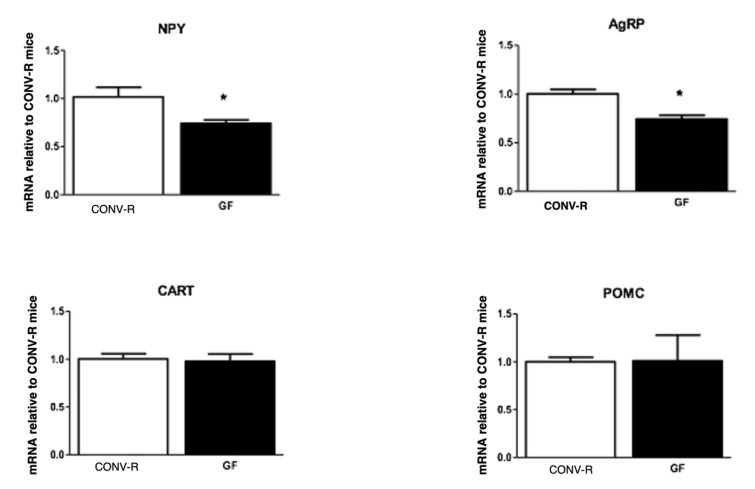
mRNA expression of NPY, AgRP, POMC, and CART in the hindbrain of GF and CONV-R mice after a 5-h fast. mRNA expression of GF mice is relative to the mean level of their CONV-R counterparts. Data are expressed as means ± SEM. * denotes statistical difference between GF and CONV-R, * *p* < 0.05.

**Figure 3 ijms-23-11756-f003:**
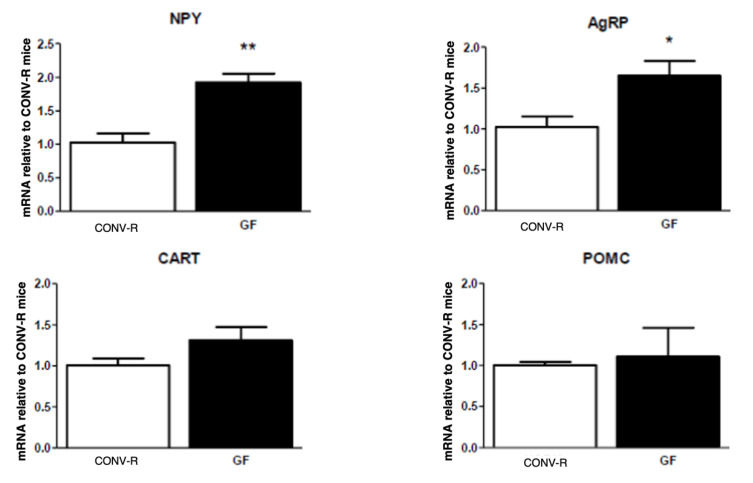
mRNA expression of NPY, AgRP, POMC, and CART in the hypothalamus of GF and CONV-R mice after a 5-h fast. mRNA expression of GF mice is relative to the mean level of their CONV-R counterparts. Data are expressed as means ± SEM. * denotes statistical difference between GF and CONV-R, * *p* < 0.05, ** *p* < 0.01.

**Figure 4 ijms-23-11756-f004:**
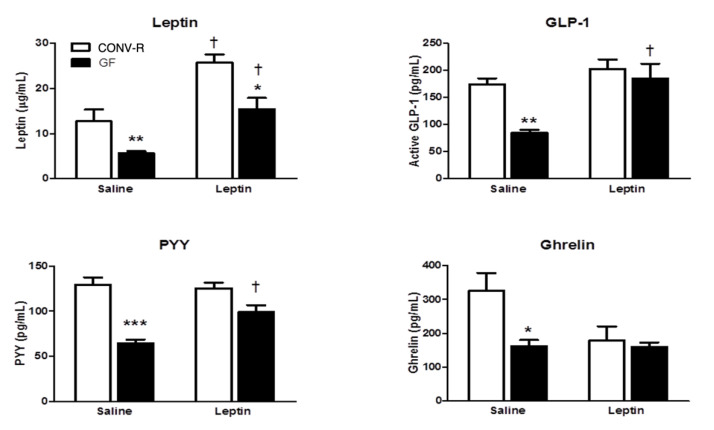
Gut peptide (leptin, GLP-1, PYY, and ghrelin) and leptin response to leptin injection in GF and CONV-R mice. Data are expressed as means ± SEM. * denotes statistical difference between GF and CONV-R. * *p* < 0.05, ** *p* < 0.01, *** *p* < 0.001. ^†^ denotes difference between phenotype and within treatment, ^†^
*p* < 0.05.

**Figure 5 ijms-23-11756-f005:**
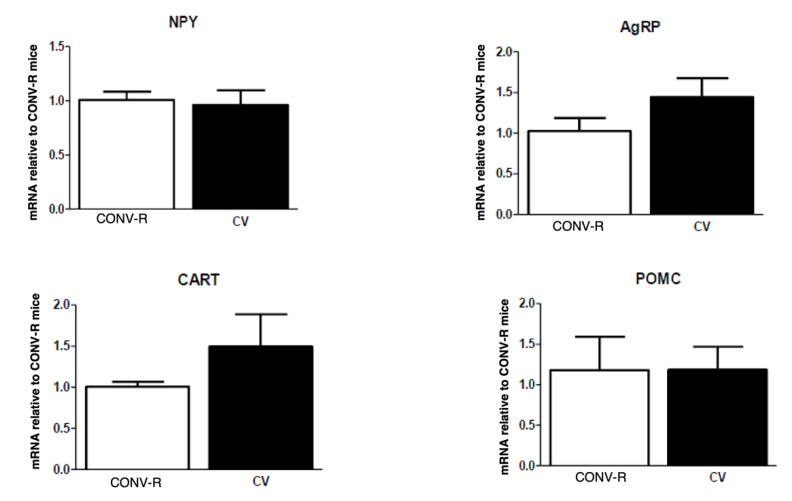
qPCR of hindbrain central peptides in 10 days conventionalized (CV) vs. CONV-R. mRNA expression levels of CV mice are relative to the mean level of CONV-R counterparts. Data are expressed as means ± SEM.

**Figure 6 ijms-23-11756-f006:**
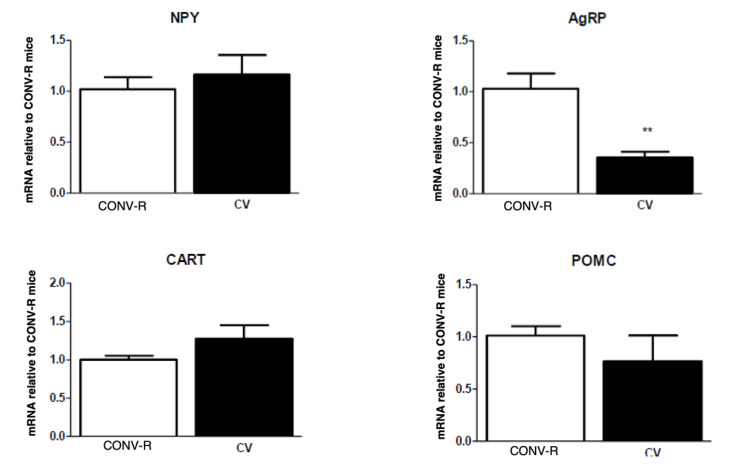
qPCR of hypothalamic peptides in 10 days conventionalized (CV) vs. CONV-R. mRNA expression levels of CV mice are relative to the mean level of CONV-R counterparts. Data are expressed as means ± SEM. ** denotes statistical difference between GF and CONV-R. ** *p* < 0.01.

## Data Availability

Data available upon request from the corresponding author.

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
