# Peer review of "Gut Microbiota Restores Central Neuropeptide Deficits in Germ-Free Mice"

_ijms, 2022, doi:10.3390/ijms231911756_

Round 1

Reviewer 1 Report

In this study, authors investigated the association between gut microbiota and centra neuropeptide deficits in GF mice.

This interesting study is well executed, explained and written. I have only a few comments:

- in Introduction, better emphasis on novelty of this study and gap in the literature that is filling could be made

-in diccussion, usually there is no sub-chapters

- Additionally, in the diccussion, better emphasis on limitations and potential clinical implications of this study could be written

Author Response

In this study, authors investigated the association between gut microbiota and centra neuropeptide deficits in GF mice.

Thank you for your constructive and positive comments which helped improve our paper.

This interesting study is well executed, explained, and written. I have only a few comments:

1) In Introduction, better emphasis on novelty of this study and gap in the literature that is filling could be made

Response: Thank you for this comment. We added more information regarding leptin and its impact on central neuropeptides. The additional information provides better background to the study and introduces the goal of our study better – which was to determine effects of gut microbiota in energy homeostasis through evaluating responses of central neuropeptides to leptin in conventionally raised versus germ free mice.

2) in discussion, usually there is no sub-chapters

Response: We have removed the sub-chapters within the discussion and re-arranged the text in a way that flows better.

3) Additionally, in the discussion, better emphasis on limitations and potential clinical implications of this study could be written

Response: Thank you for this comment. We added a conclusions section that addresses limitations, clinical implications, and future directions of this study.

Reviewer 2 Report

Hamamah and Covasa here present a work about the impact of gut microbiota on appetite regulation through the hypothalamus and hindbrain regions. The general idea of the manuscript is interesting and might give insights in the role, these microbiota might play e.g. in adipositas. However, I have one major and some minor points to adress:

Major: the manuscript is somehow fragmentary. Why did the authors not investigate both, leptin and gut peptides in plasma as well as neuronal peptides in the two brain regions in all three experimental settings? This would have given a much more complete insight. E.g., what happens with hypothalamic NPY expression following leptin administration etc.? Does the here used set-up of conventionalization leads to normalization of leptin levels...There is no rational for not doing this as animals were sacrificed anyway.

Minor points:

I would suggest to call mice with a normal gut microbiota not "normal" but conventionally raised.

Some references are lacking in the introduction ( line 60,line 68).

The paragraph of the introduction line 60 to 66 is not clear for me. How can a decreased SCFA production be observed in GF mice with further GPR41 knockout? Shoulnd't there be no microbial SCFA production at all - independent from genotype?

Why were only male mice used? What is the rationale for the 5h fasting and the 17h (!) repeated fasting for leptin injectons? At least the latter must be really exhausting for the mice. Why had this to be done? Age of the mice should be indicated besides weight.

Why was food intake not assessed?

Time point of sacrifice should be mentioned.

Where conventionalized mice also fasted for 5 h before sacrifice? What does "exposure to normal laboratory conditions" mean?

Some information about vendors is missing in methods section (e.g. for leptin). Concentration of e.g. aprotinin should be given.

Describe anesthesia.

Is it sure that beta-actin does not respond to leptin?

Conventionalization of mice is not demonstrated as e.g. measuring CFU from feces on Schaedler agar or any other state of the art method.

It is not usual to describe results within the figure legends, e.g. line 149.

Axis titles are missing (Figure 2,3,5).

I am not sure why the authors talk about increased leptin sensitivity in GF mice after exogenous administration of leptin. For me, this looks quite like a normalization of circulating leptin and by this a normalization of other gut peptides.

How can the authors conlude that leptin has a higher influence on ghreelin as the microbiota (line 345)? 

Author Response

Hamamah and Covasa here present a work about the impact of gut microbiota on appetite regulation through the hypothalamus and hindbrain regions. The general idea of the manuscript is interesting and might give insights in the role, these microbiota might play e.g. in adiposities. However, I have one major and some minor points to address:

Thank you for your constructive and positive comments which helped improve our paper.

Major: the manuscript is somehow fragmentary. Why did the authors not investigate both, leptin and gut peptides in plasma as well as neuronal peptides in the two brain regions in all three experimental settings? This would have given a much more complete insight. E.g., what happens with hypothalamic NPY expression following leptin administration etc.? Does the here used set-up of conventionalization leads to normalization of leptin levels...There is no rational for not doing this as animals were sacrificed anyway.

Response: Thank you for your insightful and legitimate comment. The main objective of the study was to 1) examine if there were any changes in brain neuropeptides in germ free mice compared to conventionally raised animals and 2) to assess whether the reverse was true (i.e if restoration of gut microbiota restored those changes); In addition we tested systemic leptin, GLP-1. PYY and ghrelin responses in animals. In as much as many papers over the years documented the role of leptin and subserving neural substrates on food intake and energy balance in regular, conventionally raised animals, some of which have been cited in this paper, we concur with the reviewer that examining these changes in all our experimental settings might have strengthened the overall findings. Notwithstanding, our paper clearly showed that there were significant differences in the hypothalamic and brain neuropeptides involved in the energy balance between GF and CONV-R animals and that bacterial colonization restored those changes. In addition, our paper clearly showed that GF mice respond differently to exogenous leptin challenge, emphasizing the role of gut microbiota on gut and brain peptides controlling caloric intake.  We have made these points clearer in the Introduction as well as discussed, the limitations.

Minor points:

1) I would suggest to call mice with a normal gut microbiota not "normal" but conventionally raised.

Response: We have changed normal (NORM) mice to conventionally raised mice with the abbreviation (CONV-R) throughout the text as recommended and remade the figures to change NORM to CONV-R to be consistent throughout the text.

2) Some references are lacking in the introduction (line 60, line 68).

Response: We added references at the end of the sentences in lines 60 (Reference 9 and 11) and at the end of line 68 (Reference 14)

3) The paragraph of the introduction line 60 to 66 is not clear for me. How can a decreased SCFA production be observed in GF mice with further GPR41 knockout? Shouldn’t there be no microbial SCFA production at all - independent from genotype?

Response: Thank you for noticing this, we removed “decreased SCFA production” and changed the sentence to read more clearly.

4) Why were only male mice used? What is the rationale for the 5h fasting and the 17h (!) repeated fasting for leptin injections? At least the latter must be really exhausting for the mice. Why had this to be done? Age of the mice should be indicated besides weight.

Response: We only used male to avoid variability due to the influence/changes of sex steroid hormones that play a fundamental role on the control of energy homeostasis and the neurotransmitters studied. The 5-hour fast was to normalize any changes that might be due to the differential responses to food intake. Rodents are nocturnal eaters, thus fasting between12-17-hour is commonly used in most studies examining the contribution of neural/metabolic signals on food intake. Further, and specific to our studies, it is known that fasting, through mainly a leptin-dependent process, induces large increases in arcuate nucleus NPY and AgRP expression which reflects modulation of the action potential of NPY/AgRP neurons. Hence, we fasted both groups to induce the threshold of the respective neuronal response through expression of the measured neuropeptides.

The age of animals has been added.

5) Why was food intake not assessed?

Response: This is an excellent point, and, in fact, we considered recording food intake in our initial protocol. However, from our extensive experience with GF animal studies, it is quite challenging to measure food intake in the isolators, particularly in mice, given the significant spillage that occur, and we were not set up for electronic food recordings inside the isolators. Therefore, we were not confident in the accuracy of the food intake data given the high food spillage and relatively small sample. We added a conclusions section with a few paragraphs addressing limitations of the study at the request of Reviewer #1 and mentioned that we did not assess food intake.

6) Time point of sacrifice should be mentioned.

Response: The mice were sacrificed after the 17 hour fast (1700 – 1000) and 2 hours after the leptin injection so at 1200 (noon). We added that they were sacrificed at 1200 in the text. The mice undergoing a 5 hour fast (0900 – 1400) were sacrificed at 1400.

7) Where conventionalized mice also fasted for 5 h before sacrifice? What does "exposure to normal laboratory conditions" mean?

Response: Yes, conventionalized mice were also fasted for 5 hours before sacrificed. Exposure to normal laboratory conditions means they were housed outside of the isolators. We added text in the manuscript to clarify both points.

8) Some information about vendors is missing in methods section (e.g. for leptin). Concentration of e.g. aprotinin should be given.

Response:

This information has been added.

9) Describe anesthesia.

Response:

Mice were sacrificed via decapitation. Our apologies for the inadvertent error.

10) Is it sure that beta-actin does not respond to leptin?

Response: There is no evidence that beta-actin responds to leptin. In addition, the data were normalized relative to conventionally raised animals.

11) Conventionalization of mice is not demonstrated as e.g.measuring CFU from feces on Schaedler agar or any other state of the art method.

Response: For conventionalization, GF mice were removed from the isolator and exposed to normal laboratory conditions. Several studies used this method of conventionalization where conventionally raised mice and germ-free mice are taken out of isolation and kept together where they are exposed to the environment and can undergo coprophagy (PMID: 32387495, 33070663, 26586378). This method of conventionalization has been found to be effective and is used regularly. In addition, colonization was confirmed via fecal sample culture tests. This was added to Methods section.

12) It is not usual to describe results within the figure legends, e.g. line 149.

Response: We have removed the descriptions of the results in the figure legends and added to the results sections when applicable.

13) Axis titles are missing (Figure 2,3,5).

Response: We have added the axis title to figures 2, 3, 5 and 6. We have also changed the NORM to CONV-R in the figure axes as well to address comment #1.

14) I am not sure why the authors talk about increased leptin sensitivity in GF mice after exogenous administration of leptin. For me, this looks quite like a normalization of circulating leptin and by this a normalization of other gut peptides.

Response: Where appropriate, we’ve changed “increased leptin sensitivity” to “increased leptin response” in GF mice after exogenous administration of leptin, compared to CONV-R animals. We agree with the reviewer’s line of thought that, at least, in acute setting, there seems to be a normalization of circulating leptin and subsequent responses of other gut peptides. This was added to the discussion section.

15) How can the authors conclude that leptin has a higher influence on ghrelin as the microbiota (line 345)? 

Response: Thank you for this comment. Our results show that leptin reduces ghrelin levels in conventionally raised mice only while leptin injection does not change ghrelin levels in germ free mice. Therefore, we have changed the sentences to say that this suggests that leptin reduces ghrelin levels through processes that may involve the gut microbiota

Round 2

Reviewer 2 Report

I would like to thank the authors for thoroughly revising the manuscript. I am still convinced that the manuscript would have benefitted from a complete analysis of all parameters in all settings, however, I can now see the point they are focussing on and agree with it.

In the improved revised version of the manuscript, some language mistakes need to be corrected such as:

Line 77, 78: “Conversely, starvation and low leptin levels, increases the drive ..” must be changed to “Conversely, starvation and low leptin levels increase…”

Line 83 “On the other hand, the hormone inhibits…” “On the other hand” should be used together with a “on one hand”.

Line 89: the term “modified gut models” is not clear to me.

Line 92 “in GF animals, especially leptin, which has been found to be significantly reduced.” ..the role of leptin?

Line 96: “NPY, AgRP, POMC and“ change to „NPY, AgRP, POMC, and”

Line 101 “a well-known peptide for its effects on hypothalamic” change to “a peptide well-known for its effects on hypothalamic”

Line 478 “This is in acord w” change to “This is in accordance w”

Line 491 “Additionally, our study evaluate the” change to “Additionally, our study evaluates/evaluated the”

Author Response

In the improved revised version of the manuscript, some language mistakes need to be corrected such as:

Thank you again for your assistance with improving our paper.  We have made all corrections, as recommended.

Line 77, 78: “Conversely, starvation and low leptin levels, increases the drive ..” must be changed to “Conversely, starvation and low leptin levels increase…”

Response: We have made the change as recommended.

Line 83 “On the other hand, the hormone inhibits…” “On the other hand” should be used together with a “on one hand”.

Response: We added “On one hand,” on the sentence before as recommended.

Line 89: the term “modified gut models” is not clear to me.

Response: We have clarified this by changing modified gut models to “high-fat induced obesity models”.

Line 92 “in GF animals, especially leptin, which has been found to be significantly reduced.” ..the role of leptin?

Response: We have changed this to the “role of leptin” as recommended.

Line 96: “NPY, AgRP, POMC and“ change to „NPY, AgRP, POMC, and”

Response: We have added the comma as recommended.

Line 101 “a well-known peptide for its effects on hypothalamic” change to “a peptide well-known for its effects on hypothalamic”

Response: We have made the change as recommended.

Line 478 “This is in acord w” change to “This is in accordance w”

Response: We have made the change as recommended.

Line 491 “Additionally, our study evaluate the” change to “Additionally, our study evaluates/evaluated the”

Response: We have made the change as recommended.